# Prolyl Hydroxylase 3 Knockdown Accelerates *VHL*-Mutant Kidney Cancer Growth In Vivo

**DOI:** 10.3390/ijms22062849

**Published:** 2021-03-11

**Authors:** Niki M. Zacharias, Lei Wang, Tapati Maity, Li Li, Steven W. Millward, Jose A. Karam, Christopher G. Wood, Neema Navai

**Affiliations:** 1Department of Urology, University of Texas MD Anderson Cancer Center, Houston, TX 77030, USA; nmzacharias@mdanderson.org (N.M.Z.); lei.gy.wang@gmail.com (L.W.); tmaity@mdanderson.org (T.M.); lli8@mdanderson.org (L.L.); jakaram@mdanderson.org (J.A.K.); cgwood@mdanderson.org (C.G.W.); 2Department of Cancer Systems Imaging, University of Texas MD Anderson Cancer Center, Houston, TX 77030, USA; smillward@mdanderson.org; 3Department of Translational Molecular Pathology, University of Texas MD Anderson Cancer Center, Houston, TX 77030, USA

**Keywords:** prolyl hydroxylases, renal cell carcinoma, hypoxia inducible factor 2α

## Abstract

Von Hippel Lindau (VHL) inactivation, which is common in clear cell renal cell carcinoma (ccRCC), leads directly to the disruption of oxygen homoeostasis. VHL works through hypoxia-inducible factors (HIFs). Within this VHL-HIF system, prolyl hydroxylases (PHDs) are the intermediary proteins that initiate the degradation of HIFs. PHD isoform 3′s (PHD3) role in ccRCC growth in vivo is poorly understood. Using viral transduction, we knocked down the expression of PHD3 in the human ccRCC cell line UMRC3. Compared with control cells transduced with scrambled vector (UMRC3-SC cells), PHD3-knockdown cells (UMRC3-PHD3KD cells) showed increased cell invasion, tumor growth, and response to sunitinib. PHD3 knockdown reduced HIF2α expression and increased phosphorylated epidermal growth factor (EGFR) expression in untreated tumor models. However, following sunitinib treatment, expression of HIF2α and phosphorylated EGFR were equivalent in both PHD3 knockdown and control tumors. PHD3 knockdown changed the overall redox state of the cell as seen by the increased concentration of glutathione in PHD3 knockdown tumors relative to control tumors. UMRC3-PHD3KD cells had increased proliferation in cell culture when grown in the presence of hydrogen peroxide compared to UMRC3-SC control cells. Our findings illustrate (1) the variable effect of PHD3 on HIF2α expression, (2) an inverse relationship between PHD3 expression and tumor growth in ccRCC animal models, and (3) the role of PHD3 in maintaining the redox state of UMRC3 cells and their proliferative rate under oxidative stress.

## 1. Introduction

About 90% of kidney cancers are renal cell carcinomas (RCCs) [1], the incidence of which has been increasing over the past decade [2]. Despite recent advances in therapy, the mean survival duration of patients with metastatic RCC is only 13–27 months [1]. The development of effective therapies to improve these patients’ dismal prognoses requires a better understanding of the mechanisms underlying the carcinogenesis of RCC [3].

Clear cell RCC (ccRCC) accounts for approximately 70% of all RCCs and metabolic reprogramming plays a central role in ccRCC carcinogenesis. Approximately 90% of ccRCCs show inactivation of the von Hippel Lindau gene, *VHL* [1]. *VHL* inactivation increases the expression levels of hypoxia-inducible factors (HIFs), which drive ccRCC carcinogenesis. Expression of HIFα, which is oxygen-labile, and HIFβ, which is constitutively expressed, leads to the transcription of multiple genes associated with angiogenesis, cell survival, and glycolysis [4,5,6]. Of the 3 known HIFα isoforms, HIF1α and HIF2α are most important in the carcinogenesis of ccRCC [7]. In the presence of oxygen, HIF proteins are hydroxylated on proline residues, which provides a recognition site for E3 ubiquitin ligases such as von Hippel Lindau protein. Ubiquitination of HIF proteins leads to their subsequent degradation by the proteasome. Hydroxylation of HIF proteins is facilitated by prolyl hydroxylases (PHDs). Under normoxic conditions, all 3 PHD isoforms (PHD1, PHD2, and PHD3) hydroxylate conserved proline residues in HIFα. However, the activity and expression pattern of PHD3 is distinct from PHD1 and PHD2 [8]. PHD3 is more active towards HIF2α than HIF1α and retains its activity even under hypoxic conditions.

PHD3 expression is downregulated in prostate and breast cancer cell lines but upregulated in RCC, some squamous cell carcinomas, and glioblastomas [8]. Given its role in HIF regulation, PHD3 expression could play a significant role in ccRCC progression. The purpose of this study was to determine the role of PHD3 in ccRCC growth in cell culture and animal models.

## 2. Results

### 2.1. RCC Cell Lines Have Varying Levels of PHD3 Expression

We carried out RT-PCR analysis to measure the level of PHD transcripts in RCC cell lines. This analysis revealed A-498, UMRC3, A-704, and 786-O cells had higher PHD3 expression levels than the 5 other RCC cell lines (Figure 1a,b). All cell lines had similar PHD1 expression levels but had varying PHD2, HIF1α, HIF2α, and VHL expression levels. Based on our previous experience with UMRC3 cells in culture [9], and A498 being an outlier in the PHD3 expression pattern, we chose to perform shRNA knockdown of PHD3 in UMRC3 cells in order to interrogate its effect on cell proliferation, migration, and metabolism. UMRC3 cells transfected with PHD3 shRNA (UMRC3-PHD3KD) showed reduced PHD3 protein expression compared with UMRC3 cells transfected with scrambled shRNA (UMRC3-SC cells) (Figure 2a).

### 2.2. PHD3 Knockdown Increases Cell Migration but Not Cell Proliferation

We next sought to understand the relationship between PHD3 expression and UMRC3 cell migration in culture. Cell migration was measured using wound healing assays. Complete wound closure was observed within 24 h with UMRC3-PHD3KD cells, while wound closure was incomplete at 30 h with UMRC3-SC cells (Figure 2b). UMRC3-PHD3KD cells also showed enhanced mobility over 12 h relative to UMRC3-SC cells in cell migration assays (Figure 2c). Interestingly, there was no significant difference in the proliferation rate of UMRC3-PHD3KD cells relative to UMRC3-SC cells at 24, 48, or 72 h in culture (Figure 2d). Similar results were observed in 786-O cells where PHD3 expression was reduced by shRNA knockdown (Figure A1).

### 2.3. PHD3 Knockdown Increases Tumor Growth

To determine the effects of PHD3 suppression on tumor growth, 5 million UMRC3-PHD3KD or UMRC3-SC cells were injected into the right flank of SCID mice in 3 separate experiments. In all experiments, the growth rate of the UMRC3-PHD3KD tumors was significantly higher than UMRC3-SC tumors. UMRC3-PHD3KD tumors were significantly larger than UMRC3-SC control tumors 21 and 28 days after cell injection (Figure 3a, *p* < 0.02 (21 days) and *p* < 0.0005 (28 days) with 8 to 10 animals per measurement). Spider plots for individual animals presented in Figure A2.

### 2.4. PHD3 Knockdown Sensitizes Tumors to Sunitinib

Sunitinib is an orally available multi-target tyrosine kinase receptor inhibitor (TKI) that is a first line therapy for ccRCC patients [10]. One of its main targets are vascular endothelial growth factor (VEGF) receptors resulting in the disruption of angiogenesis signaling. In RCC, HIF2α can play a major role in driving VEGF receptor expression [11]. To determine how PHD3 knockdown interacts with sunitinib treatment, UMRC3-PHD3KD and UMRC3-SC tumor-bearing mice were treated daily with sunitinib after tumor volumes reached approximately 100 mm^3^. Untreated UMRC3-PHD3KD tumors (NT) were significantly larger than sunitinib-treated UMRC3-PHD3KD tumors (TR) (Figure 3b, *p* < 0.0001 with 3 to 8 animals per measurement). A larger size reduction is observed in UMRC3-PHD3KD tumors after 12 and 16 days of sunitinib treatment compared to sunitinib-treated UMRC3-SC tumors (Figure 3c, *p* < 0.002, 4 to 8 animals per measurement). Spider plots for individual animals are provides in Figure A2.

### 2.5. Varying HIF2α and pEGFR Expression with PHD3 Knockdown and Sunitinib Treatment

To determine the correlation between PHD3, HIF1α, and HIF2α expression, Western analysis was performed on lysates from untreated tumors (NT) and tumors treated with sunitinib (TR) (Figure 4a). Untreated UMRC3-PHD3KD tumors had lower HIF2α expression than untreated UMRC3-SC tumors (*p* < 0.002 with 3 to 5 replicates per tumor type). However, following treatment both tumor types had equivalent HIF2α expression (Table 1). PHD3 expression was statistically lower in UMRC3-PHD3 TR tumors compared to UMRC3-SC TR tumors (*p* < 0.001 with 8 to 10 replicates per tumor type).

In glioblastoma cell lines, PHD3 increases the levels of phosphorylated epidermal growth factor receptor (EGFR) by inhibiting receptor internalization [12,13]. To determine if pEGFR upregulation is associated with PHD3 loss in UMRC3, we performed Western analyses on tumor lysates (Figure 4b). We observed a statistically significant increase in pEGFR levels in untreated UMRC3-PHD3KD tumors versus untreated UMRC3-SC tumors (*p* < 0.05, 4 replicates per tumor type). Equivalent expression of total EGFR was observed in both tumor types (Figure 4c). pEGFR levels were found to be equivalent in PHD3 knockdown and scrambled tumors following sunitinib treatment.

### 2.6. PHD3 Knockdown Changes Cellular Redox

Prior work revealed that metabolism varies after PHD3 knockdown in cell culture models of RCC [14]. However, using nuclear magnetic resonance (NMR) spectroscopy metabolomic profiling, we observed no significant differences in the levels of lactate and other metabolites between UMRC3-PHD3KD tumors and UMRC3-SC tumors (2-way ANOVA with multiple comparisons, Table A1). However, the levels of glutamine, glutathione and the spectral resonance for trimethylamine are different between the 2 tumor types suggesting that PHD3 expression modulates glutamine metabolism and oxidative stress. Therefore, we measured the expression of multiple proteins associated with glutamine metabolism (Figure 4d–f). We observed suppressed expression of ASCT2 in both untreated and treated UMRC3-PHD3KD tumors compared to UMRC3-SC tumors (Figure 4d, *p* < 0.01, 3 to 4 replicates per tumor type). Sunitinib-treated UMRC3-PHD3KD and UMRC3-SC tumors showed lower GLS1 expression relative to their untreated counterparts, while a slight reduction in GLS1 expression was observed between untreated UMRC3-PHD3KD tumors compared to untreated UMRC3-SC tumors (Figure 4e). Conversely, GLS2 expression, based on densitometry measurements, is similar in UMRC3-PHD3KD and UMRC3-SC tumors and does not change after sunitinib treatment (Figure 4f).

To determine how PHD3 knockdown affects cellular redox, we determined the concentration of reduced glutathione (GSH) and oxidized glutathione (GSSG) in UMRC3-PHD3KD versus UMRC3-SC tumors. A colorimeteric assay for glutathione showed that UMRC3-PHD3KD tumors had twice the amount of GSSG and GSH relative to UMRC3-SC tumors (Figure 5a, data are from 2 separate experiments, 8 samples per experiment, * *p* < 0.005, Holm-Sidak multiple t-test method). The GSH:GSSG ratio for UMRC3-PHD3KD tumors was smaller than that for UMRC3-SC tumors (2.6 versus 3.2 respectively), suggesting increased oxidative stress in the PHD3KD tumors [15]. To further explore the relationship between reactive oxygen species and PHD3 expression, we measured the amount of hydrogen peroxide (H_2_O_2_) produced by UMRC3-PHD3KD and UMRC3-SC cells. Bioluminescence assays revealed that UMRC3-PHD3KD and UMRC3-SC cells produced different concentrations of H_2_O_2_ (Figure 5b, 8 to 6 replicates per condition, *p* < 0.002). In addition, we observed higher proliferation of UMRC3-PHD3KD versus UMRC3-SC cells in the presence of H_2_O_2_ (Figure 5c). Initially, cells were grown in high concentrations of H_2_O_2_ (156 µM to 5 mM). In both cell types, few cells were observed with concentrations of H_2_O_2_ higher than 312.5 µM (six replicates per condition). However, higher proliferation was observed with UMRC3-PHD3KD versus UMRC3-SC cells with H_2_O_2_ concentrations between 25 µM to 200 µM (one-way ANOVA with Tukey multiple comparisons, data are from 2 separate experiments, 12 replicates per condition, *p* < 0.05).

## 3. Discussion

We observed varying levels of HIF1α, HIF2α, PHD3, and VHL expression among 9 RCC cell lines, which supports previous studies showing that RCC depends on hypoxia pathways [1,4,6,16]. The knockdown of PHD3 in the UMRC3 ccRCC cell line led to increased tumor growth in mouse models which was inhibited by sunitinib treatment. This increased growth rate with PHD3 knockdown is counterintuitive if PHD3′s primary role is the degradation of HIF proteins. These results are in contrast to Miikkulainen et al., who reported that the siRNA-mediated knockdown of PHD3 reduces the proliferation and colony formation of the ccRCC cell lines 786-O and RCC4 [14]. We found no difference in the proliferation rates of UMRC3-PHD3KD and UMRC-SC cells in culture; however, higher proliferation in UMRC3-PHD3KD than UMRC3-SC cells occurs when cell lines are grown in the presence of hydrogen peroxide. Both cell types had different rates of wound healing and migration. Western blot analysis of proteins involved in hypoxia and metabolism pathways revealed differential expression of HIF2α, GLS1, pEGFR, and ASCT2 between UMRC3-PHD3 KD and UMRC3-SC tumors. We believe these differences lead to increases in ROS-mediated growth in UMRC3-PHD3 KD tumors and cells [17].

We found that PHD3 knockdown reduced HIF2α expression in cell culture and animal models. Like those of Miikkulainen et al. [18], our results show that HIF2α and PHD3 expression are correlated in ccRCC. Untreated UMRC3-PHD3KD tumors had lower levels of HIF2α expression than untreated UMRC3-SC tumors. However, both tumor types had equivalent HIF2α expression after sunitinib treatment. The mechanistic reason for this change is unclear. However, prospective and retrospective studies have shown that HIF2α and sunitinib response are correlated. Depending on the study, HIF2α expression either improves response and overall survival [19,20] or is correlated with resistance [16]. The effects of HIF2α expression on sunitinib response may depend on context. For example, in treatment-naïve ccRCC, low HIF2α expression could drive sunitinib response. Conversely, over time, low HIF2α expression could play a role in sunitinib resistance.

Our findings illustrate that PHD3′s ability to affect HIF2α expression is variable and that a reduced level of HIF2α does not necessarily correlate with reduced tumor growth. HIF2α’s downstream and upstream interactions in the hypoxia pathway are important for understanding the overall mechanism of growth inhibition and could inform the clinical development of HIF2α inhibitors such as PT2399. A small-molecule HIF2α antagonist, PT2399, inhibits ccRCC growth in humans and animal models [21,22,23,24]. PT2385, an analog of PT2399, is in a phase II clinical trial for the treatment of patients with von Hippel-Lindau disease-associated ccRCC (NCT03108066). Cho et al. found that the sensitivity of RCC to PT2399 is highly dependent on HIF2α levels but that certain RCC cell lines (UMRC-2, 769-P, and SKRC-20) are insensitive to the drug both in culture and animal models, even though all 3 cell lines have HIF2α expression and mutant VHL [23]. In addition, Chen et al. found that sensitivity to PT2399 varied among patient-derived xenograft models of ccRCC [22]. Mutant VHL, HIF2α, and p53 clearly interact and are involved in ccRCC carcinogenesis, but their precise roles in tumor repression and growth remain unclear [23].

Our findings on the role of PHD3 in tumor growth are in good agreement with those seen in glioblastoma by the Acker laboratory [12,13]. They found the loss of PHD3 in glioma cells increased EGFR signaling, increased proliferation and reduced apoptosis relative to PHD3-expressing cells under low growth factor and starvation conditions. This function of PHD3 was found to be independent of its proline hydroxylation activity. In addition, this group observed increased hypermethylation of the promoter for the gene (*EGLN3*) for PHD3 in glioblastoma tumors compared to normal brain. They concluded that PHD3 knockdown is a direct control mechanism to allow cancer cells to sustain tumor growth even under normally toxic conditions such as hypoxia. Therefore, loss of PHD3 could enhance survival in ccRCC by facilitating accelerated growth in the hypoxic environment of the kidney. Indeed, we observed increased levels of pEGFR in UMRC3-PHD3KD tumors compared to scrambled controls which was effectively eliminated following sunitinib treatment. In addition, *EGLN3* methylation is significantly different in kidney cancer tissue versus normal tissue (Figure A3, [25]) suggesting that epigenetics might play a role in PHD3 expression in ccRCC.

microRNA has also been shown to regulate PHD3 expression [26]. MicroRNA-1205 (miR-1205) promoted cell proliferation and increased the resistance to H_2_O_2_ induced apoptosis in castration resistant prostate cancer cell lines through its post-transcriptional regulation of the *EGLN3* gene. This effect was found to be mediated by a direct interaction between miR-1205 with *EGLN3* mRNA leading to reduced expression of PHD3. Overexpressing PHD3 in prostate cells expressing miR-1205 reduced proliferation and resistance to H_2_O_2_ [26].

One of the most striking effects of PHD3 knockdown in UMRC3 cells is the effect on intracellular glutathione concentrations. We observe twice the amount of GSH and GSSG in UMRC3-PHD3KD tumors versus UMRC3-SC tumors. We propose that this increased concentration of glutathione allows UMRC3-PHD3KD cells to adapt to high concentrations of reactive oxygen species (ROS). As such, we see higher proliferation of UMRC3-PHD3KD cells grown in the presence of H_2_O_2_ at concentrations between 25 µM to 200 µM compared to UMRC3-SC cells. We hypothesize that PHD3 knockdown increases ROS-mediated proliferation. However, further experiments are needed to confirm this hypothesis.

We also observe differences in the expression of glutaminolysis proteins following PHD3 knockdown. We found that ASCT2 (also named SLC1A5), a major glutamine transporter [27], was suppressed in both untreated and sunitinib-treated PHD3-knockdown tumors. Our data suggests that PHD3 may have a role in the regulation of this transporter and in glutamine metabolism overall. Glutamine metabolism is upregulated in ccRCC [28,29] and a link between HIF2α and ASCT2 has been reported before [30]. However, in our experiments, ASCT2 expression remained low in sunitinib-treated PHD3 knockdown tumors, whereas HIF2α expression was equivalent between sunitinib-treated PHD3-knockdown tumors and control tumors. After glutamine is transported into a cell, it can be metabolized through several pathways, including those fueling the citric acid cycle [31]. In the mitochondria, GLS1 converts glutamine to glutamate. We found that PHD3-knockdown and control tumors had different levels of GLS1 expression, but this difference was not as pronounced as that in ASCT2 expression between the 2 tumor types. The cytoplasmic GLS2 enzyme expression is similar in sunitinib-treated and untreated UMRC3-PHD3KD and UMRC3-SC tumors. Our data suggests that PHD3 expression affects ASCT2 expression; however, the mechanism underlying this effect is unclear. In other cancers, ASCT2 expression can promote tumor growth [32,33,34,35]. However, fourteen known transporters can transport glutamine [36]; and glutamine transporter plasticity and redundancy have been observed in many different cancer cells [37,38,39]. Additional experiments will be required to further elucidate changes in the expression pattern of glutamine transporters following PHD3 knockdown.

Our results provide an improved understanding of PHD3 and HIF2α expression in RCC. With the generation of HIF2α and glutaminolysis inhibitors, it is vital to understand the manner in which hypoxia pathways are regulated in a VHL mutant cancer such as ccRCC. Indeed, the components of this pathway may provide prognostic biomarkers to identify patient responders. Our experiments should be repeated in other ccRCC cell lines to establish the role of PHD3 in other genetic backgrounds. Deletion of the *EGLN3* gene using CRISPR/Cas9, further interrogation of the PHD3/EGFR signaling axis, and evaluation of the effects of PHD3 knockdown on metastatic progression in orthotopic ccRCC animal models could further clarify PHD3′s role in this disease.

## 4. Materials and Methods

### 4.1. Cell Culture

The human RCC cell lines A-498, 786-O, ACHN, 769-P, A704, and Caki1 were obtained from ATCC (Manassas, VA, USA), whereas the cell lines SN12C [40] and UMRC3 [9] were developed in our institution. UMRC3, A-498, 786-O, A704, and ACHN cells were grown in minimal essential medium; 769-P cells were grown in RPMI-1640 medium; Caki1 cells were grown in McCoy’s 5A medium; and SN12C cells were grown in high-glucose Dulbecco’s modified Eagle medium. All cell lines were grown in the presence of 10% fetal bovine serum, 300 mg/mL streptomycin, 100 U/mL penicillin, and 1 non-essential amino acids. Short tandem repeat analysis was used to authenticate the cell lines. Cells were detached using 0.25% trypsin with 2.21 mM EDTA. Most of the work uses human ccRCC cell line UMRC3 [9]. This cell line was derived directly from the primary kidney tumor of a patient with metastatic ccRCC. This cell line does have a VHL mutation [41] and injection into a nude or SCID mouse leads to tumor formation and metastatic progression.

### 4.2. Reverse Transcription Polymerase Chain Reaction

Total RNA was isolated from cell pellets using the mirVana miRNA Isolation Kit (Thermo Fisher Scientific, Waltham, MA, USA) according to manufacturer’s instructions. RNA was quantified using ultraviolet spectrophotometry. cDNA was synthesized from 1 µg of RNA using the ThermoScript reverse transcription polymerase chain reaction (RT-PCR) system and Platinum Taq DNA polymerase with Oligo (dT) primers (all from Thermo Fisher Scientific). The following primers were used for RT-PCR: LIMD1: forward 5′AAGGATGGGCTCTTCCGAGT3′, reverse 5′TCTCCATGAGTGAGGCAGGA3′; PHD1: forward 5′GAAAAAGCTCGCCACCCTG3′, reverse 5′AGGGAGAGCCTGACTTAGGG3′; PHD2: forward 5′AGCCCAGTTTGCTGACATTGAA3′, reverse 5′ACTTTAGCTCGTGCTCTCTCAT3′; PHD3: forward 5′GTGGCTTCCCATCCCCAAAA3′, reverse 5′CAGGAAGTTGTCCAGGTAGCA3′; HIF1α: forward 5′AGGAGGATCACCCTCTTCGT3′, reverse 5′CTCCATGGTGAATCGGTCCC3′; HIF2α: forward 5′GTACAATCCTCGGCAGTGTC3′, reverse 5′GACCCGAAAAGAGGACGGAG3′; VHL: forward 5′GAGATGCAGGGACACACGAT3′, reverse 5′ATCCGTTGATGTGCAATGCG3′; and GAPDH: forward 5′CTCCTGTTCGACAGTCAGCC3′, reverse 5′TTCCCGTTCTCAGCCTTGAC3′. PCRs were repeated for 35 cycles. The RT-PCR products were run on 2% agarose gel, stained with ethidium bromide, and imaged (Bio-Rad Gel-Doc System, Hercules, CA, USA).

### 4.3. Lentiviral Transfection

We used EGLN3 MISSION shRNA transduction particles (type SHCLNV-NM_022073, Millipore Sigma (Burlington, MA, USA)) to knock down PHD3 according to manufacturer’s protocol. Briefly, UMRC3 cells were seeded in 96-well plates at 5000 cells per well and grown for 24 h. The media was then changed, and the cells were infected with lentivirus in the presence of 5 µg/mL polybrene. After expansion and the addition of a selection marker (8 µg/mL puromycin), multiple clones were selected and expanded. The clones were subjected to Western blot analysis; a single clone with the lowest PHD3 expression (UMRC3-PHD3KD) was used in all experiments. Following the same protocol, we used scrambled shRNA lentiviral particles (sc-108080, Santa Cruz Biotechnology, Dallas, TX, USA) to create a single clone (UMRC3-SC), which was used as the control cell line in all experiments. Similar procedures were followed for the generation of 786-O-PHD3KD and 786-O-SC cells used in experiments represented in Figure A1.

### 4.4. Wound Healing Assay

UMRC3-SC and UMRC3-PHD3KD were plated in 6-well plates. After the cells became over confluent, a 200-µL pipette tip was used to scratch each well, and the plates were imaged under a microscope every 6 h for up to 30 h. Each well was considered a replicate (6 replicates per cell line). Similar procedures were used in wound healing assay with 786-O-PHD3KD and 786-O-SC cells. Plates were imaged under the microscope every 6 h up to 24 h. (Figure A1).

### 4.5. Migration Assay

To assess cell migration, we used a CytoSelect 24-Well Cell Invasion Assay, Basement Membrane (cat. #CBA-100, Cell Biolabs, Inc., San Diego, CA, USA) according to the manufacturer’s instructions. Briefly, 500 µL of serum-containing media was added to the lower chamber of each well, and 20,000 cells in 300 µL of serum-free media were placed in the upper chamber. After 24 h, the transwell inserts were removed and stained with CyQuant^®^ GR Dye supplied by the manufacturer. 3 wells per cell line were stained.

### 4.6. Cell Proliferation Assay

Cells were plated in a 96-well plate at 5000 cells per well. At the indicated times, the cells in 8 wells were treated with trichloroacetic acid (10% *w*/*v*), washed, and then treated with sulforhodamine B solution as described previously [42]. The 510 nm absorbance was measured per well using a plate reader (FLUOStar Omega, BMG Labtech, Cary, NC, USA).

### 4.7. Hydrogen Peroxide Measurements in Cell Culture

Cells were plated in a 96-well plate at 5000 cells in 80 µL of solution per well. For this assay, cells were plated in Gibco MEM media without phenol red and glutamine (51200038, Thermo Fisher Scientific, Waltham, MA, USA) with glutamine (Millipore Sigma Aldrich, G7513, Burlington, MA, USA) added to a final concentration of 1 mM. In addition, media contained 10% fetal bovine serum, 300 mg/mL streptomycin, 100 U/mL penicillin, and 1× non-essential amino acids. Cells were grown for 24 h. Instructions for the Promega ROS-Glo™ H_2_O_2_ assay (G8820, Promega, Madison, WI, USA) were followed. For the assay, 20 µL of substrate mix was added to each well and cells remained in CO_2_ incubator for 4 h prior to adding 100 µL of ROS-Glo detection solution, incubating for 20 min at room temperature, and measuring the bioluminescence using a plate reader (FLUOStar Omega, BMG Labtech, Cary, NC, USA).

### 4.8. Cell Proliferation Assay in the Presence of Hydrogen Peroxide

Cells were plated in a 96-well plate at 5000 cells per well. For this assay, cells were plated in Gibco MEM media without phenol red and glutamine (51200038, Thermo Fisher Scientific) with glutamine (Millipore Sigma Aldrich, G7513) added to a final concentration of 1 mM. In addition, media contained 10% fetal bovine serum, 300 mg/mL streptomycin, 100 U/mL penicillin, and 1× non-essential amino acids. Media containing H_2_O_2_ was freshly made using 30 % *w/v* H_2_O_2_ (9.8 M, Millipore Sigma Aldrich, H-1009) in the same media that was used for plating the cells. Serial dilution used to generate 10 mM, 5 mM, 2.5 mM, 1.25 mM, 800 µM, 625 µM, 400 µM, 312.5 µM, 200 µM, 156.25 µM, 100 µM, 50 µM, and 25 µM H_2_O_2_ solutions. Cells grown for 24 h and then the same amount of media (with or with H_2_O_2_) used to plate cells was added. All final concentrations were taken as half of the original H_2_O_2_ concentration. Cells were grown for another 24 h, media removed, cells washed with media and then treated with Promega Cell Titer 96 Aqueous Non-Radioactive Cell Proliferation Media (G5421, Promega). Cells sat in CO_2_ incubator for 90 min prior to determining the 490 nm absorbance using a plate reader (FLUOStar Omega, BMG Labtech).

### 4.9. Mouse Models

All animal experiments were carried out in accordance with international standards. All animal experimental procedures were approved by the University of Texas MD Anderson Institutional Animal Care and Use Committee (Protocol 1200, Approved 5/16/2017). Male and female mice were used in all experiments. Mice were purchased from MD Anderson Experimental Radiation Oncology NOD *scid* gamma mouse colony. The colony uses mating pairs from the Jackson Laboratory (Stock number 005557, Sacramento, CA, USA). We subcutaneously injected 5 million UMRC3-PHD3KD or UMRC3-SC cells in 100 µL of phosphate-buffered saline into the flanks of SCID mice. Tumors were allowed to grow for 2 weeks and then measured with calipers approximately every 4 days; tumor volume was determined using the equation (1/2) × [length × width^2^]. In 2 separate experiments after the tumors were approximately 100 mm^3^, the mice received 200 µL of 6.25 mg/mL sunitinib malate dissolved in water daily by oral gavage or were left untreated. When tumors were larger than 1.5 cm, mice bearing UMRC3-PHD3KD or UMRC3-SC subcutaneous tumors were humanely killed using anesthetic overdose and cervical dislocation and their tumors were removed. The tumor tissues were immediately flash-frozen.

### 4.10. GSH/GSSG Assay

Flash-frozen untreated UMRC3-PHD3KD and UMRC3-SC tumor tissues were homogenized in ice-cold phosphate-buffered saline using a freeze/thaw cycle in liquid nitrogen and lysis D beads (MP Biomedicals, Irvine, CA, USA). The solution was centrifuged at 4000× *g* for 5 min at 4 °C. The supernatant was transferred to a 1.5-mL centrifuge tube and then centrifuged again at 14,000 *g* for 10 min at 4 °C. We added 300 µL of the supernatant to 300 µL of ice-cold 5% SSA (*w*/*v*) solution (5-sulfo-salicylic acid dehydrate in 20 mL of water). A protease and phosphatase inhibitor cocktail (cat. #78441, Thermo Fisher Scientific) was added to the remainder of the supernatant, which was stored until further use. The protein concentrations of all lysates were determined using the Pierce BCA Protein Assay Kit (cat. #23225, Thermo Fisher Scientific). 287 µg of protein was used in each sample. SSA-treated samples were analyzed using the Glutathione Colorimetric Detection Kit (cat. #EIAGSHC, Thermo Fisher Scientific) according to the manufacturer’s protocol. Absorbance at 405 nm was measured using a CLARIOstar plate reader (BMG Labtech).

### 4.11. Nuclear Magnetic Resonance (NMR) Based Metabolomics

We wanted to understand the changes in metabolomics between UMRC3-PHD3KD and UMRC3-SC tumors. NMR spectroscopy was used to analyze metabolites in UMRC3-PHD3KD and UMRC3-SC tumors. 70 mg of tumor mass used per sample. Flash-frozen tissues were homogenized under liquid nitrogen using a mortar and pestle. Metabolites were extracted from the homogenized tissues in 3 freeze/thaw cycles using an ice-cold 2:1 methanol:water solution and 500 µL of Lysing Matrix D Beads (cat. #6933, MP Biomedicals) as described previously [43]. The resulting solution was centrifuged at 4000× *g* for 3 min. The supernatant was removed and then lyophilized to dryness, and the remaining metabolites were resuspended in 1X phosphate buffer with 10% deuterium oxide. We used 3-(trimethylsilyl)-1-propanesulfonic acid-d_6_ (cat. #613150, Millipore Sigma) as an internal standard for all NMR samples. All samples were run on 500 MHz Bruker Biospin Avance III high-definition NMR instrument equipped with a Prodigy BBO cryoprobe. One-dimensional ^1^H-NMR spectra used 256 scans, spectral width of 10,245 Hz, and water suppression performed with presaturation. Data were processed and analyzed with MestreNova (Mestrelab Research, Santiago de Compostela, Spain) and the identification of specific resonances was determined using the Human Metabolome Database (http://www.hdmb.ca, accessed on 18 January 2020) [44].

### 4.12. Western Blotting and Densitometry

Tissues or cells were lysed on ice in 1× RIPA lysis and extraction buffer (cat. #89900, Thermo Fisher Scientific) with Halt protease inhibitor (cat. #78441, Thermo Fisher Scientific). Protein concentrations were determined using the Pierce BCA Protein Assay Kit (cat. #23225, Thermo Fisher Scientific). Total protein (20 µg) was loaded onto the gels, and blots were developed with primary antibodies against PHD3 (ab184714, Abcam, Cambridge, UK; 1:2000 dilution); HIF1α (D2U3T, Cell Signaling Technology, Danvers, MA, USA; 1:1000 dilution); HIF2α (D9E3, Cell Signaling Technology; 1:1000 dilution); fumarate hydratase (ab95950, Abcam; 1:3000 dilution); EGFR (sc-373746, Santa Cruz Biotechnology, Dallas, TX; 1:1000 dilution). pEGFR (sc-57545, Santa Cruz Biotechnology; 1:200 dilution); glutaminase (GLS1; ab93434, Abcam; 1:1000 dilution), glutaminase 2 (GLS2; NBPI-76544, Novus Biologicals, Littleton, CO, USA; 1:1000 dilution); alanine-serine-cysteine transporter 2 (ASCT2; ABN73, Millipore Sigma, 1:1000 dilution), and β-actin-horseradish peroxidase (5125S or 12262S, Cell Signaling Technology; 1:5000 dilution). Image J (National Institutes of Health, Bethesda, MD, USA) or Image Studio Lite (version 5.2, LI-COR Biosciences, Lincoln, NE, USA) were used to perform densitometry.

### 4.13. Statistical Analysis

GraphPad Prism 8 software (San Diego, CA, USA) was used to perform all statistical analyses. Statistical significance between two cohorts was determined using unpaired t-test or between multiple cohorts was determined using 2-way ANOVA with Sidak’s multiple comparisons unless stated otherwise in the text. *p* values less than 0.05 were considered significant.

## Figures and Tables

**Figure 1 ijms-22-02849-f001:**
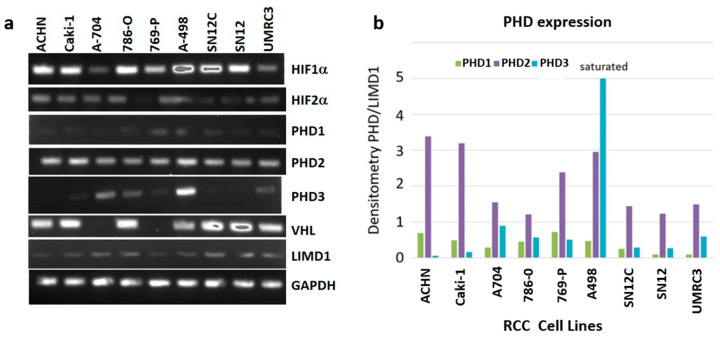
RCC cell lines show varying levels of PHD3 expression. (**a**) Agarose gel electrophoresis of RT-PCR products of RNA isolated from the indicated RCC cell lines shows varying levels of the PHD3 transcript. (**b**) Densitometry analysis of the gel shown in Figure 1a. Measurements are plotted as ratios of PHD1, PHD2, and PHD3 to the scaffold protein LIMD1. Transcript signal in the A-498 lysate was found to be saturated.

**Figure 2 ijms-22-02849-f002:**
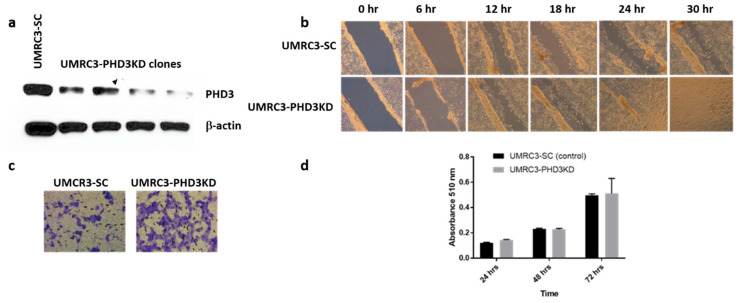
Effects of PHD3 knockdown in vitro. (**a**) Western analysis confirmed significant reduction of PHD3 expression in UMRC3 cells transfected with PHD3 shRNA (UMRC3-PHD3KD) relative to cells transfected with scrambled shRNA (UMRC3-SC). All subsequent in vitro and in vivo experiments used the clone with highest PHD3 knockdown efficiency. (**b**) Wound healing assays revealed that UMRC3-SC cells (top row) showed incomplete wound healing at 30 h, whereas UMRC3-PHD3KD cells (bottom row) showed complete wound healing within 24 h. 6 replicates per cell line were performed. (**c**) Migration assays revealed more extensive migration of UMRC3-PHD3KD cells compared to UMRC3-SC cells over a 12 h period. (**d**) Sulforhodamine B assays revealed that the proliferation rates of UMRC3-PHD3KD cells and UMRC3-SC were not significantly different at 24, 48, or 72 h (8 samples per time point per cell line).

**Figure 3 ijms-22-02849-f003:**
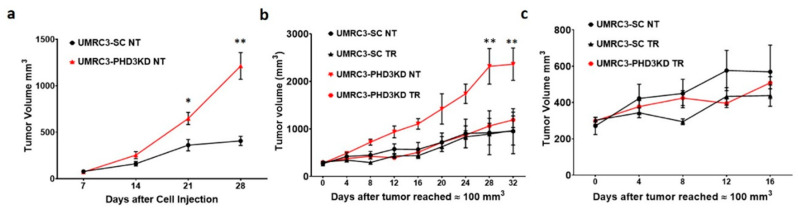
UMRC3-PHD3KD tumors grow more rapidly than UMRC3-SC tumors. (**a**) UMRC3-PHD3KD tumors were significantly larger than UMRC3-SC control tumors at 21 days (* *p* < 0.02) and 28 days (** *p* < 0.0005) after cell injection. (**b**) Untreated UMRC3-PHD3KD tumors (NT) were significantly larger than sunitinib-treated UMRC3-PHD3KD tumors (TR) throughout the experiment with the greatest significance between cohorts observed 28 and 32 days post-treatment (** *p* < 0.0001). (**c**) A replotting of the growth curves for UMRC3-SC NT, UMRC3-SC TR and UMRC3-PHD3KD TR tumors to observe the change in tumor size between UMRC3-SC NT and UMRC3-SC TR cohorts. The combination of sunitinib treatment and PHD3 knockdown resulted in a larger size reduction in UMRC3 tumors relative to sunitinib treatment alone. This was readily observable in the size difference of treated versus untreated UMRC-PHD3KD and UMRC-SC tumors at 12 and 16 days (*p* < 0.002).

**Figure 4 ijms-22-02849-f004:**
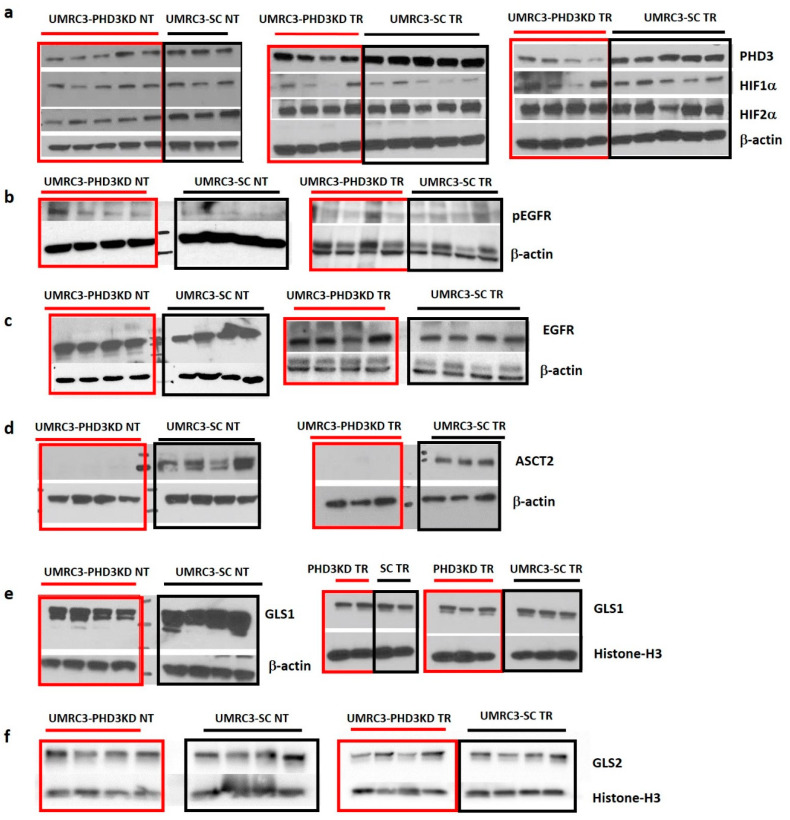
PHD3 knockdown reduced HIF2α expression and increased pEGFR expression in untreated tumors (NT). This effect disappeared after sunitinib treatment (TR). UMRC3-PHD3KD tumor samples are highlighted in red boxes while UMRC3-SC tumor samples are highlighted in black boxes. Densitometry ratio measurements of each protein over the loading control shown in Table 1. (**a**) Western analysis and densitometry revealed that untreated UMRC3-PHD3KD tumors had slightly lower PHD3 expression but significantly lower HIF2α expression (*p* < 0.002) compared to untreated UMRC3-SC tumors. Western analysis and densitometry also revealed that sunitinib-treated UMRC3-PHD3KD tumors had lower PHD3 expression (*p* < 0.001), but essentially equivalent HIF2α expression, relative to sunitinib-treated UMRC3-SC tumors. (**b**) Western analysis revealed higher pEGFR expression in untreated UMR3-PHD3KD tumors compared to untreated UMRC3-SC tumors (*p* < 0.05), which was not present after sunitinib treatment. (**c**) EGFR expression was similar between UMRC3-PHD3KD and UMRC3-SC tumors in both treated and untreated cohorts. (**d**) Western analysis showed suppressed expression of ASCT2 in UMRC3-PHD3KD tumors relative to UMRC3-SC tumors independent of sunitinib treatment. In both blots, the difference in expression of ASCT2 was significant (*p* < 0.01). (**e**) GLS1 expression is slightly lower in untreated UMRC3-PHD3KD tumors versus untreated UMRC-SC tumors. However, we observe lower overall expression of GLS1 in all tumors (UMRC3-PHD3KD and UMRC3-SC) after sunitinib treatment. No difference in GLS1 expression was observed between sunitinib-treated UMRC3-PHD3KD and UMRC3-SC tumor samples. (**f**) GLS2 expression is similar in UMRC-PHD3KD and UMRC3-SC tumor samples and is not affected by sunitinib treatment.

**Figure 5 ijms-22-02849-f005:**
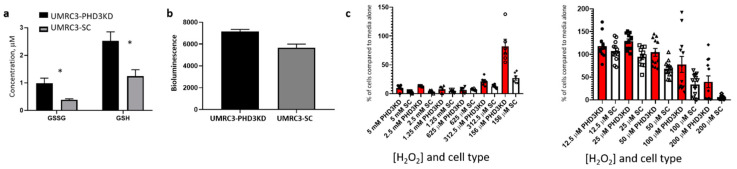
Redox changes by PHD3 knockdown. (**a**) Glutathione colorimetric detection revealed that UMRC3-PHD3KD tumors had significantly higher concentrations of GSH and GSSG than UMRC3-SC tumors (* *p* < 0.005). (**b**) UMRC3-PHD3KD and UMRC3-SC cells produce different levels of hydrogen peroxide measured by bioluminescence assay over a 24 h period (* *p* < 0.002). (**c**) UMRC3-PHD3KD cells have higher proliferation rates in the presence of H_2_O_2_ than UMRC3-SC cells. Concentration of H_2_O_2_ and cell type (PHD3KD versus SC) is shown on the in x-axis and percentage of cells in each well compared to media alone is displayed on the y-axis. In both cell types, few cells remain at H_2_O_2_ concentrations higher than 312.5 µM. Significance differences in proliferation are observed in UMRC3-PHD3KD cells (red bars) versus UMRC3-SC cells (white bars) at 200, 156, 100, 50 and 25 µM H_2_O_2_ concentrations (*p* < 0.05).

**Table 1 ijms-22-02849-t001:** Densitometry measurement of PHD3, HIF1α, HIF2α, pEGFR, and EGFR over loading control protein as visualized in Figure 4.

Tumor Type	PHD3/ β-actin	HIF1α/ β-actin	HIF2α/ β-actin	pEGFR/ β-actin	EGFR/ β-actin	ASCT2/ β-actin	GLS1/ β-actin	GLS1/ Histone-H3	GLS2/ Histone-H3
UMRC3-PHD3 NT	0.38 ± 0.15	0.40 ± 0.11	* 0.50 ± 0.15	* 0.13 ± 0.04	5.58 ± 0.57	* 0.0081 ± 0.00047	1.11 ± 0.26		1.04 ± 0.30
UMRC3-SC NT	0.59 ± 0.03	0.44 ± 0.11	* 0.95 ± 0.27	* 0.05 ± 0.03	5.32 ± 0.87	* 0.70 ± 0.32	1.50 ± 0.48		0.79 ± 0.32
UMRC3-PHD3 TR	* 0.21 ± 0.09	0.23 ± 0.19	0.92 ± 0.18	0.21 ± 0.15	0.50 ± 0.16	* 0.019 ± 0.024		0.48 ± 0.13	0.83 ± 0.51
UMRC3-SC TR	* 0.54 ± 0.06	0.25 ± 0.10	0.77 ± 0.42	0.23 ± 0.07	0.52 ± 0.13	* 0.81 ± 0.16		0.53 ± 0.07	0.77 ± 0.08

Average ± standard deviation given for each protein. NT stands for no treatment while TR stands for sunitinib treatment. * PHD3 expression was statistically lower in UMRC3-PHD3 TR tumors compared to UMRC3-SC TR tumors (* *p* < 0.001, 8 to 10 replicates per tumor type). While HIF2α expression was statistically lower in UMRC3-PHD3 NT tumors compared to UMRC3-SC NT tumors (* *p* < 0.002, 3 to 5 replicates per tumor type). pEGFR expression was found to be higher in UMRC3-PHD3KD NT tumors compared to UMRC3-SC NT tumors (* *p* < 0.05, 4 replicates per tumor type). ASCT2 expression was found to be lower between UMRC3-PHD3KD and UMRC3-SC tumors in both treated and untreated cohorts (* *p* < 0.01, 3 to 4 replicates per tumor type).

## Data Availability

The data that support the findings of this study are available from the corresponding author upon reasonable request.

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
