# Peer review of "Prolyl Hydroxylase 3 Knockdown Accelerates VHL-Mutant Kidney Cancer Growth In Vivo"

_ijms, 2021, doi:10.3390/ijms22062849_

Round 1

Reviewer 1 Report

Authors have not addressed experimentally how reduced ASCT2 expression in PHD3-deficient tumors can be still compatible with higher tumor growth and glutathione content. However the have provided a possible explanation in the discussion section.

Reviewer 2 Report

The authors have adequately addressed my concerns and questions. No further comments.

This manuscript is a resubmission of an earlier submission. The following is a list of the peer review reports and author responses from that submission.

Round 1

Reviewer 1 Report

Authors have addressed satisfactorily some - not all - of my previous comments. Therefore I suggest to address the following two points.

  1. Regarding the novel data about the role of PHD3 in cellular tolerance to oxidative stress. Authors show that PHD3-deficient tumors have higher levels glutathione content that might explain (i) this higher resistance of PHD3-silenced UMRC3 cells to oxidative stress and (ii) possibly the tumor growth advantage upon PHD3 inhibition. However the expression of the central glutamine carrier ASCT2 is severely reduced in PHD3-silenced tumors. How it can be understood higher resistance to oxidative stress with reduced levels of ASCT2 taking into consideration that glutathione biosynthesis requires glutamine uptake? Moreover how it can be understood increased in vivo tumor growth with reduced levels of ASCT2 taking into consideration its role in tumor growth (PMID: 29326164, PMID: 26455325, PMID: 28759021, PMID: 33464409)? Along this line, authors should assess ASCT2 and glutathione levels in PHD3-silenced cells and not only xenografts. This is relevant because - as mentioned in one my initial comments - molecular differences detected might be secondary to the fact that PHD3-deficient and control xenografts grow differentially. Alternatively authors might discuss how to reconcile reduced ASCT2 with higher glutathione biosynthesis and cell proliferation.

  1. In line 219, ‘conentrations’ should be corrected by ‘concentrations’. In this line authors should check the entire manuscript for possible typos.

Reviewer 2 Report

The authors have properly addressed all the concerns and no further experiments are required to support their conclusions. However, the manuscript would highly benefit  from language revision to improve clarity and soundness.
